# Prescribing and Safety of Direct-Acting Oral Anticoagulants Compared to Warfarin in Patients with Atrial Fibrillation on Chronic Hemodialysis

**DOI:** 10.3390/pharmacy8010037

**Published:** 2020-03-10

**Authors:** Estella Davis, Dallin Darais, Kevin Fuji, Paige Nekola, Khalid Bashir

**Affiliations:** 1School of Pharmacy and Health Professions, Creighton University, Omaha, NE 68178, USA; kevinfuji@creighton.edu (K.F.); paigenekola@creighton.edu (P.N.); 2Medical Center, CHI Health-Creighton University, Omaha, NE 68178, USA; Dallin.Darais2@alegent.org (D.D.); Khalid.Bashir@alegent.org (K.B.); 3Center for Health Services Research and Patient Safety, Creighton University, Omaha, NE 68178, USA; 4School of Medicine, Creighton University, Omaha, NE 68178, USA; 5Dialysis Clinics, Inc., Omaha, NE 68131, USA

**Keywords:** end-stage renal disease, ESRD, ESKD, hemodialysis, HD, direct oral anticoagulation, DOACs, atrial fibrillation, AF

## Abstract

ESRD patients receiving hemodialysis (HD) were excluded from landmark trials evaluating direct-acting oral anticoagulants (DOACs) in atrial fibrillation (AF). The objective was to evaluate prescribing and bleeding with DOACs compared to warfarin in AF patients with chronic HD. A retrospective, observational study of patients receiving warfarin or DOAC from April 2010-April 2016 from area health system hospitals and Dialysis Clinics, Inc. records. Data was analyzed using descriptive statistics, ANOVA, and chi-square. Ninety-one patients were included with warfarin as the initial OAC in most patients (n = 76) at average dose of 29 mg/week. Fifteen patients were initially prescribed apixaban (n = 12) or dabigatran (n = 3). Most switches in OAC therapy were to apixaban. When the initial OAC was a DOAC, it was not dosed appropriately in five with one bleed, two dosed appropriately had bleeds. When initial warfarin was switched to a DOAC, it was not dosed appropriately in seven with five bleeds. More bleeds occurred with warfarin alone (n = 18) vs. those on warfarin switched to DOAC (n = 5) vs. DOAC alone (n = 3), p = 0.022. All but four patients that bled had HAS-BLED scores three or higher. Warfarin was most often prescribed and associated with a higher incidence of bleeding compared to DOACs in this population of patients at high risk for bleeding. Larger studies should be conducted to analyze the impact of DOAC dose appropriateness on safety and clinical outcomes.

## 1. Introduction

Chronic kidney disease (CKD) is defined as a reduction in renal function in glomerular filtration rate (GFR) < 60 mL/min/1.73 m^2^ for three months or longer or with the presence of albuminuria. CKD can gradually progress to severe end-stage renal disease (ESRD) where GFR < 15 mL/min/1.73 m^2^ or patients require renal replacement therapies, such as hemodialysis (HD), peritoneal dialysis, or a kidney transplant. It is estimated that 37 million people in the US have CKD, affecting 38% of patients greater than 65 years of age [1]. Atrial fibrillation (AF) is also a common complication affecting up to 6.1 million people and 9% of the US population greater than 65 years of age [2]. The prevalence and incidence of CKD and AF have been increasing due to improved longevity and shared collective risk factors for both diseases, including diabetes mellitus and hypertension [3]. AF among patients with CKD is high, occurring in approximately one-quarter of the population [4]. The presence of both AF and CKD further exacerbates stroke and mortality risk with a 66% increase in the relative risk of death [3]. Large epidemiologic studies show that patients with ESRD or those requiring renal replacement therapy, are 1.8 to 2 times higher risk for stroke or systemic thromoboembolism compared to those with no kidney disease [5,6]. ESRD patients are also 2.7 times higher risk for bleeding compared to those with no kidney disease [6].

Patients with AF at high risk for stroke should be anticoagulated to prevent stroke or systemic thromboembolism [7,8]. Oral anticoagulants (OACs) available in the US to reduce the risk for developing stroke or systemic embolism in patients with AF include warfarin, dabigatran, apixaban, rivaroxaban, and edoxaban. The latter four medications are termed direct oral anticoagulants (DOACs). Compared to warfarin for the indication of AF, DOACs have similar or superior efficacy, less bleeding risk, and more predictable pharmacokinetic and pharmacodynamic parameters with fixed dosing regimens; making them an attractive alternative to prescribe over warfarin. Oral anticoagulation for the population of ESRD patients receiving HD is challenging due to the exclusion of patients with severe renal impairment or those on renal replacement therapies, including HD, from landmark efficacy and safety studies for DOACs [9,10,11,12]. At the time of this study, the 2014 US guidelines for AF only recommended warfarin as the anticoagulant of choice, did not recommend dabigatran or rivaroxaban due to lack of evidence regarding the balance of risks and benefits, and made no mention of apixaban or edoxaban [7]. The 2012 American College of Chest Physicians (CHEST) guidelines for antithrombotic therapy for AF did not give guidance regarding DOAC use in HD patients and stated further randomized controlled trials were necessary [13].

Despite the lack of guidance from US guidelines for DOAC use in AF patients receiving HD and lack of prospective trials to validate their safety and efficacy, these agents were indeed prescribed in this population. At the time of this study, the only published study evaluating prescribing patterns of oral anticoagulants (OACs) in the population of AF patients with ESRD on HD was conducted by Chan and colleagues [14]. The study found an increase in prescribing for both dabigatran and rivaroxaban in this HD population shortly after their approval for use in the US, despite the exclusion of patients with severe renal impairment or on dialysis from landmark studies and lack of evidence of their safety in HD patients. This study found a significant increase in the risk of hospitalization or death from bleeding with dabigatran and rivaroxaban compared to warfarin in the HD population. A limitation to this study was that no patients received apixaban or edoxaban. Therefore, at the time of our study, no data was available evaluating the safety or outcomes using apixaban or edoxaban in the dialysis population.

The primary objective of this observational study was to evaluate initial OAC prescribing choice, DOAC dose selection, and changes to OAC therapies for AF in patients with chronic, intermittent ESRD receiving HD. Secondary objectives were to evaluate adverse events including the incidence of stroke or thromboembolic events and incidence of bleeding events including major bleeding or clinically relevant non-major bleeding (CRNMB), which could include dialysis-access site bleeding.

## 2. Materials and Methods

A retrospective, observational study was conducted to evaluate a cohort of patients with AF who received at least one dose of an OAC while hospitalized at a Catholic Health Initiatives (CHI) medical center in the Omaha, NE metro area between 20 April 2010 to 30 April 2016 who then continued to receive an OAC while receiving chronic outpatient HD at one of four metro area Dialysis Clinics, Incorporated (DCI) facilities under the care of a CHI nephrologist. Patients were included for evaluation if they were >19 years of age and met both criteria of ESRD requiring chronic outpatient HD and DOAC therapy for the diagnosis of AF. Patients with acute kidney injury (AKI) requiring HD for <3 months were excluded. The electronic medical record was evaluated to determine administration of any of the following OACs while hospitalized: warfarin, dabigatran, rivaroxaban, apixaban, or edoxaban. The hospital electronic health record or dialysis clinic database were subsequently evaluated to determine initial prescribing and changes to OAC therapy over the study time period. Patients were evaluated from the time they met inclusion criteria and followed through OAC discontinuation or the study period end date.

Demographic data collected included age, gender, race/ethnicity, weight, height, serum creatinine, creatinine clearance using the Cockcroft-Gault equation [15], documented history or diagnosis of AF, documented history of CKD with ESRD on HD, HD schedule, HD location, data to calculate CHA_2_DS_2_-VASc risk score [16] or documentation of a score, and data to calculate HAS-BLED score [17]. Anticoagulation data collected included initial OAC, dose, subsequent changes to anticoagulation or other dose adjustments, and international normalized ratios if on warfarin. A subtherapeutic range for the INR was defined as an INR value greater than 0.1 difference from the goal INR range. For example, if the goal INR range was 2–3, an INR value of ≤1.8 would be considered subtherapeutic, and an INR value of ≥3.2 would be considered supratherapeutic. An INR value between 1.9 to 3.1 would be considered therapeutic. For each patient the percent of values in subtherapeutic range (SubR%), therapeutic range (TR%), and supratherapeutic range (SupraR%) were calculated based on the number of INR values that were either subtherapeutic, therapeutic, or supratherapeutic divided by the total number of INRs documented for that patient, multiplied by 100 to get the SubR%, TR%, and SupraR%, respectively.

Appropriateness of dose for DOAC therapy was based on product labeling recommendations for HD patients at the time of the study (see Appendix A). Dosing was either termed “Dosed Appropriately = DA” or “Not Dosed Appropriately = NDA”. HD patients prescribed dabigatran, rivaroxaban, or edoxaban would be assigned “NDA” regardless of dose, as product labeling at the time of the study did not include dosing information for HD patients (dabigatran and rivaroxaban) or the product labeling expressly stated its use in HD patients is not recommended (edoxaban) [18,19,20]. Evaluation of appropriateness of dose for patients prescribed apixaban was evaluated based on whether dose reduction criteria were met [21]. Apixaban is dosed at a standard 5 mg twice daily and reduced to a lower 2.5 mg twice daily dose when two of three dose reduction criteria are met (SCr ≥ 1.5 mg/dL, weight < 60 kg, age > 80 years of age). If an apixaban patient received: (a) standard high dose 5 mg twice daily with two of three dose reduction criteria met = NDA, dose high, or (b) low dose 2.5 mg twice daily with less than two of three dose reduction criteria met = NDA, dose reduced.

Adverse event data collected included identification of hemorrhagic or ischemic stroke or thromboembolic events documented in the electronic health record. Major bleeding was defined using ISTH and RE-LY major bleeding criteria [9,22]. Therefore, major bleeding encompassed the following criteria: fatal bleeding, bleeding in a critical area or organ (i.e., intracranial bleed), bleeding causing a reduction in the hemoglobin level of at least 2 g/dL, or leading to a transfusion of at least 2 units of blood, or bleeding requiring inotropic agents or necessitating surgery. All other bleeding was considered clinically relevant non-major bleeding (CRNMB) or minor bleeding (CRNMB/minor), such as epistaxis, urological, gastrointestinal, and documented dialysis access site bleeding.

The study was considered exempt and approved by Creighton University’s IRB (approval project number 990104-1) and Dialysis Clinic Inc.’s Administrative Review Office (approval project number 2016.62). Data was analyzed using descriptive statistics. Comparisons of binary variables was assessed using chi square test or Fisher’s exact test. Continuous variables were analyzed by Student’s t test or analysis of variance (ANOVA). A *p*-value of less than 0.05 was considered significant.

## 3. Results

A total of 173 patient charts were screened for potential inclusion. Eighty-two patients were excluded because they were receiving an OAC for an indication other than AF, did not meet both criteria of ESRD-HD and OAC therapy during the study time period, or were not a chronic HD patient.

Ninety-one patients met inclusion criteria, with 76 (83.5%) initially receiving warfarin and 15 (16.5%) receiving a DOAC. The majority of patients were male, Caucasian, with an average age of 69 years of age. On average, patients were at high risk for the development of stroke and high risk for bleeding, with an average CHA_2_DS_2_-VASc score of 4.6 and HAS-BLED score of 3.8, respectively. There was no difference between the initial OAC groups of warfarin and DOAC for any demographic variables (see Table 1). The average duration of OAC therapy during the time period was 16.6 ± 17.6 months, 17.1 ± 18.5 months for warfarin, and 8.9 ± 8.0 months for DOAC therapy.

### OAC Prescribing

Warfarin was the initial OAC therapy prescribed for most patients (n = 76, 83.5%) (see Figure 1). The average weekly dose for those prescribed warfarin was 29 mg with an average TR of 41.8%, SubR of 24.4%, and SupraR of 33.9%. DOAC therapy was the initial OAC therapy for 15 patients (16.5%). Of those, seven patients were initially prescribed low dose apixaban, five received standard dose, and three received dabigatran. No patients were prescribed rivaroxaban or edoxaban. When warfarin was the initial OAC, there were nine switches to a DOAC; with two to dabigatran and seven to apixaban. The majority of switches in OAC therapy were to apixaban 5 mg or 2.5 mg (n = 12).

Twenty-six patients (28.5%) experienced a bleed and more bleeds occurred in patients on warfarin alone (n = 18) compared to those who received both warfarin and a DOAC (n = 5), or a DOAC only (n = 3) p = 0.022 (see Table 2). Most major bleeds (n = 6), CRNMB/minor (n = 10), and major plus CRNMB/minor (n = 2) occurred with warfarin alone. One patient experienced a major plus CRNMB/minor bleed after initial warfarin was switched to dabigatran 150 mg twice daily (NDA) and experienced CRNMB/minor bleed, then switched back to warfarin and experienced a major GI bleed, sepsis and death (see Table 2, Patient #19).

There were no obvious prescribing trends found from analyzing DOAC dose appropriateness and its impact on the development of bleeding events. As expected, some patients experienced a bleed when the DOAC dabigatran should not have been prescribed (see Table 2), whereas two patients initially prescribed low dose dabigatran 75 mg twice daily NDA that switched to apixaban standard dose 5 mg twice daily DA did not experience a bleed. As expected, some patients did not experience a bleed when low dose apixaban 2.5 mg twice daily NDA (dose reduced) was used, but some experienced a bleed despite using low dose apixaban (see Table 2).

The only trend identified when analyzing patient characteristics associated with bleeding events was that patients were typically at high risk for developing a bleed. Of the patients who experienced a major bleed with warfarin only (Table 2, Patient #1 through #18), the SupraT% ranged from 14–43%, only two patients were on concomitant antiplatelet aspirin therapy, but all patients were high risk of bleed with HAS-BLED scores greater than 3. Of the patients who experienced a major bleed with warfarin switched to DOAC (Table 2, Patient #19 through #23), the SupraT% ranged from 0–67%, all but one patient was on concomitant aspirin therapy, and all were at high risk for bleeding with HAS-BLED scores greater than 3.

No patients experienced a stroke or systemic thromboembolic event during the follow-up time period. One patient initially on warfarin with CRNMB/minor HD access site bleed, was switched to dabigatran 150 mg twice daily and experienced CRNMB/minor GI bleed, then switched back to warfarin eventually expired when readmitted to the hospital due to sepsis and bleeding complications (Table 2, Patient # 19).

## 4. Discussion

This small, observational study in patients with AF and ESRD receiving chronic HD illustrated that warfarin was the OAC most often initially prescribed for patients, followed by apixaban, then dabigatran. No patients were prescribed rivaroxaban or edoxaban during the study time frame. The pattern of DOAC medication prescribing from our study is similar to other DOAC choice prescribing patterns found in AF patients on HD noted in the literature showing warfarin prescribed most often (89%), followed by apixaban (9–10%), rivaroxaban (0.8–1%), and dabigatran (0.3–1%) [23,24]. Apixaban had been the only DOAC with product labeling recommendations that included dosing in dialysis patients during our study time period and that of other larger cohort studies. Therefore, it was anticipated that this DOAC would be prescribed most often. The updated 2019 US guidelines for AF differ from the 2014 guidelines to now include a recommendation for prescribing either apixaban or warfarin, with a class IIb recommendation, as anticoagulants for use in patients with an elevated risk for stroke (CHA_2_DS_2_-VASc score of at least 2 in those with end-stage CKD (CrCl < 15 mL/min) or are on dialysis [8]. The 2019 US AF guidelines were also updated to reflect a class III recommendation to avoid dabigatran, rivaroxaban, or edoxaban due to the lack of evidence from clinical trials showing the benefit exceeds the risk of these medications for patients ESRD or on HD [8]. The updated 2018 CHEST guidelines for AF continue to recommend warfarin for oral anticoagulation and include a separate remark that DOACs should generally not be used, but apixaban has dosing information for use in patients receiving HD [25]. After our study time period, small, prospective studies evaluating pharmacokinetic and pharmacodynamic parameters in healthy patients compared to those receiving a 3–4 hour HD session after single dose rivaroxaban or apixaban were published [26,27]. These pharmacokinetic studies influenced wording in revised product labeling of rivaroxaban (August 2016) and apixaban (July 2016) to include verbiage that for patients maintained on intermittent HD, administration of apixaban at the usually recommended dose or rivaroxaban at 15 mg once daily will result in concentrations of the drug and pharmacodynamic activity similar to those observed in the landmark AF trials for both agents [19,21]. Despite this labeling revision for rivaroxaban, a reduction in prescribing is expected due to results from the Chan and colleagues’ study and the class III recommendation by US AF guidelines to avoid its use in patients receiving HD [8,14].

Almost one-third of patients (n = 26) in our study experienced a bleed, with significantly more occurring with warfarin alone compared to warfarin switched to a DOAC or DOAC alone. Other studies evaluating apixaban in HD patients confirm their lower risk for bleeding compared to warfarin in AF patients [23,28]. There was significant variability in dose appropriateness when prescribing DOACs but no clear association with dose appropriateness and bleeding. Patients who experienced a bleed also had a number of other factors predisposing to bleeding, including a high proportion of supratherapeutic INRs or concomitant antiplatelet therapy. The risk of bleeding is higher in CKD and ESRD patients and the HAS-BLED score is consistently high in the CKD populations and those receiving HD [23,29]. Pathophysiological causes of increased hemorrhagic events are multifactorial including uremia related platelet dysfunction or impaired platelet adhesion and aggregation. Extrinsically, the propensity to bleed is increased due to concurrent antiplatelets, NSAIDs, frequent invasive diagnostic and invasive strategies such as central venous access and hemodialysis (with potential for frequent heparin exposure) [3]. Our study examined initiation of DOAC from the hospital setting to the outpatient dialysis setting. Acuity of illness, stopping/starting/initiation of OAC therapy, or other hospital-based factors could have impacted the risk for bleeding in our population.

After our study was complete, results from a large cohort study examining standard and low dose apixaban compared to warfarin in patients with AF on HD was published by Siontis and colleagues [23]. They found that 5 mg twice daily had significantly lower risk of stroke or systemic embolism (stroke/SE), major bleeding, and death compared to warfarin; while reduced dose apixaban 2.5 mg twice daily had lower risk of major bleeding, but no difference in stroke/SE or death. prospective studies evaluating DOAC use in AF patients on HD have preliminary results [30] or are enrolling patients [31]. Preliminary results from the prospective RENAL-AF study that randomized patients to apixaban (n = 82) or warfarin (n = 72) were recently reported at the 2019 American Heart Association meeting [30]. Most patients received standard dose apixaban at 5 mg twice daily (71.4%), and apixaban was dose was reduced from 5 mg to 2.5 mg twice daily in 27.3% of patients. There were similar rates of bleeding and stroke for those on apixaban compared to warfarin, and a numerically higher rate of all cause death with apixaban compared warfarin (25.6% vs. 18.1%). Another prospective study is underway to compare patients randomized to reduced dose apixaban 2.5 mg twice daily compared to warfarin in AF patient on HD called the AXADIA-AFNET 8 study [31]. Final results from both of these studies should provide more robust evidence evaluating the clinical efficacy and safety of standard compared to reduced dose apixaban in the HD population.

Our study attempted to examine dose appropriateness and development of a bleed; however, no obvious trend was identified beyond patients being at high risk for bleeding. There is one study that examined appropriateness of DOAC dose in AF patients receiving standard or reduced dosing that evaluated safety and clinical outcomes [29]. The study further divided the population into patients with or without a renal indication for dose reduction based on eGFR cutoff of <30 mL/min/1.73 m^2^ for dabigatran, <50 mL/min/1.73 m^2^ for rivaroxaban, and SCr ≥ 1.5 mg/dL for apixaban. This study was not specific for ESRD patients on HD and only 2.2% of the study population had eGFR < 30 mL/min/1.73 m^2^. They found that DOACs were DA in 84%, 12% were NDA (dose reduced), and 4% were NDA (dose high). Among patients that were NDA (dose high), this was associated with an over 2-fold higher risk of major bleeding and no significant difference in stroke/SE. Among patients that were NDA (dose reduced), there was an almost 5-fold higher risk of stroke/SE and no significant difference in major bleeding.

Limitations to our study include its small sample size as the study was not powered to detect a difference in safety or clinical outcomes. The retrospective study design was dependent upon documentation in the chart regarding specifics about the OAC therapy, lab values, patient characteristics influencing CHA_2_DS_2_-VASc or HAS-BLED score, or outcomes such as bleeding events. Other limitations include the evaluation period for subjects included during phases ranging from acute inpatient hospital stays to outpatient clinic visits where variability in acuity of illness could impact bleeding risk, lab values, or other outcomes. Selection of DOAC, initial or subsequent dose selection, and warfarin dosing and monitoring were left at the discretion of providers as no DOAC or warfarin dosing protocols were used. These factors could have led to wide variability in prescribing and management of OACs.

Education has been provided to pharmacists, nephrologists, cardiologists and dialysis center staff of appropriate DOAC prescribing and dose selection as there was much variability evidenced by our study results and other published studies. These groups should have increased awareness of the higher risk for bleeding in the ESRD HD population and recognize other patient factors including HAS-BLED score, history of bleeding events with previous antithrombotic therapy, and concomitant antiplatelet therapy when selecting and dosing DOAC or warfarin therapy.

Results from our study and other large studies suggest that apixaban has a lower incidence of bleeding compared to warfarin when used in AF patients on HD. Other studies suggest standard dose apixaban 5 mg twice daily is more effective than and safe than warfarin in this population, however our study was too small to determine the impact of dose selection on outcomes. It will continue to be important to examine clinical and safety outcomes from any prospective, randomized, controlled trials examining standard or reduced doses of DOACs to provide more guidance for their use in the population of AF patients on HD.

## Figures and Tables

**Figure 1 pharmacy-08-00037-f001:**
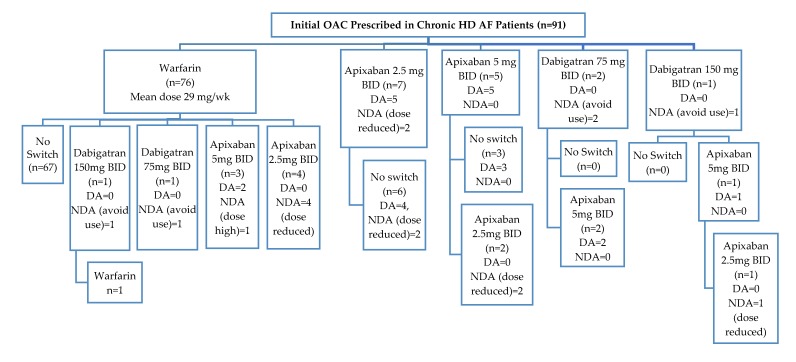
Dose Appropriateness of Initial OAC Prescribed with, Subsequent Changes to OAC Therapy. DA = Dosed Appropriately and NDA = Not Dosed Appropriately.

**Table 1 pharmacy-08-00037-t001:** Patient Demographics at the Time of Initial OAC.

Characteristic	Overall (n = 91)	Warfarin (n = 76)	DOAC (n = 15)	*p*-Value
Age (years), mean ± SD	69 ± 10.7	68.3 ± 11.0	73 ± 8.4	0.116
Gender (n), %: Male	47 (52%)	40 (53%)	7 (47%)	0.673
Ethnicity (n), %: Caucasian	72 (79%)	60 (79%)	12 (80%)	0.617
Weight (kg), mean ± SD	91 ± 23.5	90.3 ± 23.5	92.6± 4.3	0.730
SCr (mg/dL), mean ± SD	5.3 ± 2	5.4 ± 2.1	4.9 ± 1.3	0.447
CrCl (mL/min), mean ± SD	17.6 ± 8.5	17.7 ± 8.6	17.4 ± 8.2	0.903
CHA_2_DS_2_VASc score, mean ± SD	4.6 ± 1.5	4.6 ± 1.4	4.4 ± 1.9	0.650
HAS-BLED score, mean ± SD	3.8 ± 1.1	3.9 ± 1.1	3.6 ± 1.1	0.404

**Table 2 pharmacy-08-00037-t002:** Case Series Summary of Bleeding Events.

Patient Number	OAC at Time of Bleed	Cumulative Duration of OAC	DOAC Dosed Appropriately?	HAS-BLED Score	Concurrent Antiplatelet Medication	Warfarin % Values in Subtherapeutic Range (SubR%), Therapeutic Range (TR%), Supratherapeutic Range (SupraR%)			Major Bleed: Type of Bleed, INR (If Documented), Other Comments	CRNMB/Minor Bleed: Type of Bleed, INR (If Documented), Other Comments
						SubR%	TR%	SupraR%		
1	Warfarin	6 mos	-	4	ASA 81 mg	40	20	40	GI bleed, INR 1.2	-
2	Warfarin	13 mos	-	4	None	23	46	31	-	GI bleed ×2
3	Warfarin	54 mos	-	3	None	41	37	22	-	GI bleed
3	Warfarin	12 mos	-	2	None	10	63	27	-	GI bleed
5	Warfarin	44 mos	-	4	None	68	27	5	-	Epistaxis ×2
6	Warfarin	21 mos	-	3	None	32	32	36	Hematoma arm requiring surgical intervention (Transfuse 2U PRBC), INR 5.6	Epistaxis ×4
7	Warfarin	20 mos	-	3	None	18	40	42	Hematoma leg—(Transfuse 4U PRBC, Reduction Hgb ≥ 2 g/dL) GI bleed—(Transfuse 4U PRBC, Reduction Hgb ≥ 2 g/dL) Hemoptysis—(Transfuse 2U PRBC)	-
8	Warfarin	49 mos	-	6	ASA 81 mg	35	42	23	-	Hemoptysis
9	Warfarin	7 mos	-	4	None	31	26	43	Urologic bleed—(Transfuse 2U PRBC, Reduction Hgb ≥ 2 g/dL)	-
10	Warfarin	5 mos	-	2	ASA 81 mg	6	50	44	-	HD access site ×2
11	Warfarin	48 mos	-	4	ASA 81 mg	40	60	0	-	Epistaxis
12	Warfarin	43 mos	-	4	ASA 81 mg, Clopidogrel 75 mg	50	50	0	-	HD access site, Urologic
13	Warfarin	8 mos	-	3	ASA 81 mg	58	16	26	-	Hemoptysis, INR 7.8
14	Warfarin	73 mos	-	3	None	32	47	21	GI bleed—(Transfuse 3U PRBC)	-
15	Warfarin	9 mos	-	2	None	22	57	21	GI bleed—(Transfuse 2U PRBC), INR > 8.8	HD access site, Hematoma
16	Warfarin	4 mos	-	4	ASA 325 mg	29	57	14	GI bleed—(Transfuse 2U PRBC, Reduction Hgb ≥ 2 g/dL), INR 7.1	-
17	Warfarin	10 mos	-	5	ASA 81 mg	2	50	48	-	Epistaxis
18	Warfarin	8 mos	-	3	None	13	54	33	GI bleed—(Transfuse 2U PRBC, Reduction Hgb ≥ 2 g/dL)	-
19	Warfarin	1 month	-	5	ASA 81 mg	36	24	40	-	HD access site
	Dabigatran 150 BID	1 month	No, avoid use	*See above*	ASA 81 mg	-	-	-	-	GI bleed, Bruising
	Warfarin	5 mos	-	*See above*	ASA 81 mg	*See above*	*See above*	*See above*	GI bleed, Death/Sepsis	-
20	Warfarin	52 mos	-	3	None	77	10	13	-	-
	Dabigatran 75 BID	1 month	No, avoid use	*See above*	None	-	-	-	GI bleed-(Transfuse 2U PRBC, Reduction Hgb ≥ 2 g/dL)	-
21	Warfarin	1 month	-	4	ASA 81 mg	0	33	67	GI bleed-(Transfuse 4U PRBC, Reduction Hgb ≥ 2 g/dL)	-
	Apixaban 2.5 BID	2 mos	No, dose reduced	*See above*	ASA 81 mg	-	-	-	-	-
22	Warfarin	47 mos	-	3	ASA 81 mg	37	58	5	-	HD access site ×2
	Apixaban 2.5 BID	14 mos	No, dose reduced	*See above*	None	-	-	-	-	-
23	Warfarin	2 mos	-	5	ASA 81 mg	67	33	0	-	-
	Apixaban 2.5 BID	1 month	No, dose reduced	*See above*	ASA 81 mg	-	-	-	GI bleed (Reduction Hgb ≥ 2 g/dL)	-
24	Dabigatran 150 BID	7 days	No, avoid use	5	ASA 81 mg	-	-	-	-	GI bleed
	Apixaban 5 BID	15 days	Yes	*See above*	ASA 81 mg	-	-	-	GI bleed (Transfuse 2U PRBC, Reduction Hgb ≥ 2 g/dL)	-
	Apixaban 2.5 BID	24 mos	No, dose reduced	*See above*	None	-	-	-	-	-
25	Apixaban 2.5 BID	9 mos	Yes	4	ASA 81 mg	-	-	-	-	HD access site
26	Apixaban 2.5 BID	2 mos	Yes	2	None	-	-	-	-	GI bleed

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
