# Peer review of "Prescribing and Safety of Direct-Acting Oral Anticoagulants Compared to Warfarin in Patients with Atrial Fibrillation on Chronic Hemodialysis"

_pharmacy, 2020, doi:10.3390/pharmacy8010037_

Round 1
Reviewer 1 Report
Thank you for the opportunity to review this manuscript describing the prescribing of oral anticoagulants (OACs) in a small cohort of patients with AF and ESRD receiving haemodialysis, and their clinical outcomes. This is a very interesting topic, and clearly represents an important clinical issue for prescribers working largely in an evidence vacuum. Nonetheless, I believe that the manuscript would benefit from some review to optimise its relevance and readability for the interested reader.
Introduction- Overall, the Introduction provides a useful context for the research in terms of the relative importance of stroke prophylaxis in patients with CKD, and the challenges of using DOACs in patients on HD, but is overly long and detailed.
- In particular, I would recommend that the paragraph beginning at line 58 could be trimmed right back. This paragraph could be summarised in a table either in the Materials and Methods section (as the definitions of ‘Appropriate prescribing’), or as an Appendix.
- The paragraph beginning at line 83 could be summarised in 3-4 sentences – it in unnecessary to describe the conflicting recommendations of all of the major guidelines.
- The study by Chan et al. described in the paragraph beginning at line 106 is highly relevant but again should be summarised, not described in depth. A more comprehensive discussion of the main findings of this study, and how they compare to those of the current study, should be reserved for the Discussion section. Furthermore, the presence in the literature of a large retrospective cohort study (Chan et al.) plus additional observational cohort studies (Siontis et al. and Yao et al.) mean that the authors need to carefully consider, and then clearly enunciate in the manuscript, the rationale for the current study. What does this study add to the literature over and above the currently available published studies? This is currently unclear, especially given the relatively small cohort in the current study.
Materials and Methods
- The Materials and Methods are well-described and would allow the study to be replicated. Please provide references for Cockcroft-Gault equation (line 150), CHA2DS2-VASc and HAS-BLED (line 152) and the ISTH criteria (line 157).
- Please ensure that you define ‘appropriate dosing’ in this section, perhaps as a table as suggested above.
- A p value of LESS THAN 0.05 is usually considered statistically significant (line 169).
Results
- Please review the demographic results in the paragraph beginning at line 175. Most of these are repeated in Table 1, so the text can be substantially reduced.
- Please specify in the title for Table 1 (or a footnote) that these demographics are based on the patients’ initial OAC. Why are there no values for mean durations of therapy for warfarin and DOACs in Table 1?
- The TTR of the patients taking warfarin initially was quite low (41.8%). What were the mean proportions of time spent in the supra-therapeutic and sub-therapeutic ranges? This will obviously have affected the patients’ risk of bleeding.
- Figure 1 is highly descriptive and very useful. Much of the text in the paragraph commencing at line 187 can be summarised as it repeats what it presented in the figure. Furthermore, the number of initial DOAC users was very low so the actual numbers are more meaningful than the percentages presented.
- Please clarify the statement that ‘where a DOAC was initially prescribed, it was dose appropriately based on labeling recommendations available at the time of the study…’. My interpretation of the information provided in the Introduction is that the labeling of apixaban and rivaroxaban was revised sometime after 2016 (the pharmacokinetic and pharmacodynamics studies do not appear to have been published until 2016). Were there actually any labelling recommendations for these agents in patients undergoing HD at the time they were prescribed in many/any of these patients?
- Could the ‘dose appropriateness’ (appropriate/potential overdose/potential underdose) data be incorporated into Figure 1? It is a bit confusing in the text due the number of variants described.
- Line 219: Please note that the p value of 0.156 suggests that there was no statistically significant difference between the HAS-BLED scores. It is therefore not appropriate to state, ‘… the average HAS-BLED scores were higher…’
- I would suggest that Tables 2 and 3 be combined, and reformatted. Table 2 is currently confusing – the information about which drugs were associated with the bleeding events could be included in the body of the table for better clarity. It would also be worthwhile considering including other information relevant to the bleeding episodes – e.g. dosing appropriateness of the DOACs, TTR for patients taking warfarin, INR at the time of the bleed, as well as bleeding site. It may even be worth considering treating these patients as a case series (as there are only 26 patients who experienced a bleed) then the table could include one line per patient detailing severity and site of bleed(s), OAC at the time, dose appropriateness for the DOAC, INR/TTR, etc… This would allow some of the results in the text to be shortened.
- If the presentation of the results is changed, this will need to be reflected in the Abstract.
Discussion
- The Discussion would benefit from some reformatting. The first paragraph should be a summary of the overall findings, which are then discussed in more depth in the ensuing paragraphs.
- Ensure that the results are not repeated in the Discussion. They can be referred to in a general sense, but not directly repeated (e.g. line 265-271). Overall, the Discussion is very long and ‘results heavy’, including data from other studies. I would recommend revising these to only provide comprehensive results when necessary to prove a point.
- Please review the statement in lines 343-344, ‘… our small observational study and other large cohort studies show that DOACs had less incidence of bleeding that those on warfarin’ – is this true for the study by Chan et al.?
- Please include a paragraph describing the limitations of this study, including the limited generalisability due to the small number of patients taking DOACs (in particular).
- The concluding paragraph (lines 343-354) is good, but needs to be ‘tighter’ to leave the reader with a strong take home message.
Minor editorial suggestions:
- Consider concluding the Abstract with a stronger conclusion/clinical recommendation.
- Avoid commencing sentences with numbers (e.g. line 20 in the Abstract).
- Please review line 22 in the Abstract – is there a dash missing between ‘DOACs’ and ‘apixaban’?
- Please ensure consistency and accuracy in the use of the abbreviation ‘CHA2DS2-VASc’ throughout the manuscript.
- Please review the references for consistent formatting (e.g. capitalisation in ref 17).
Reviewer 2 Report
This is a carefully although small study. Results are described in a satisfactory and comprehensive way.
A few points of improvement are:
*L6 there is a typo: place of "and"
*Line 53 The risk of gastrointestinal bleeding may be higher for certain DOACs this should be taken up in the reporting and analysis
*Line 133 ISTH abbreviations should be defined when they appear
*Line 91 the quantitatieve reporting of percentages with 3 significant numbers when the population is < 100 makes not much sense. In many parts of the manuscript it would be helpfully use the real numbers (e.g. 5 of the 8 patients on .. had .... instead of %)
*Line 188 the average time within the therapeutic range is rather low, this should be discussed. In general a discussion of how anticoagulation was handled on for example HD days would be informative
*Line 126 the reader is not really interested in the timing of these studies but comparing results is of greater importance. A recent paper is highly relevant and should probably be discussed in the manuscript:
https://www.ncbi.nlm.nih.gov/pubmed/31906546
Comparison of this sample with other studies should probably also be part of the abstract and conclusion.
*Inappropriate dosing has been observed with DOAC in the settings. A brief comparison would be interesting. What steps are proposed to improve on this situation? This could be part of the conclusion.
*The following question should be addressed : Are these data important in assessing the quality of care of a dialysis centre?
*some other potential interesting questions to tackle
Are international guidelines relevant for this discussion, are product labels different in Europe of elsewhere?
Round 2
Reviewer 1 Report
Thank you to the authors for their comprehensive consideration of my previous comments. The Abstract, Introduction and Methods are now much tighter, although I still believe that the Results and Discussion are overly long and detailed, especially for a small cohort study that was underpowered to detect any significant associations with bleeding.
- Line 65: Please add ‘in’ between ‘use’ and ‘AF’ – i.e. “Despite the lack of guidance from US guidelines for DOAC use in AF patients receiving HD…"
- Lines 69-79 remain overly detailed and repetitive. Time period, methods, results, etc. of other studies should only be described in general terms, unless the details are required for comparative purposes. This also applies to the Discussion.
- Lines 110-112: Thank you for including relevant definitions in your Methods. Please note that, unless the patients’ INR results were all spaced evenly apart in terms of timing, what is reported is not TIME in therapeutic range (which is typically calculated using Rosendaal’s interpolation method), but the percentage of therapeutic/subtherapeutic/supratherapeutic INRs.
- Lines 120-126 are not required, having been defined in Appendix A.
- Lines 130-133: The definitions of major bleeding provided are somewhat contradictory. Was the threshold ‘bleeding causing a reduction in the hemoglobin level of at least 2 g/dL’ or ‘5g/dL’? And was it bleeding ‘leading to a transfusion of at least 2 units of blood’ or ‘at least 4 units of blood’? Please clarify.
- Lines 159-171: Figure 1 is highly descriptive; the results in the text are largely repeated from the figure and could be further summarised or deleted.
- Figure 1: How is ‘Med stopped’ DA? The Figure 1 title block (lines 178-9) is repetitive.
- Table 2 is well constructed and highly descriptive. Given that column 2 is ‘OAC at time of bleed’, why do some lines (e.g. Patients 21, 22, etc.) have rows where there was no bleed? If the aim is to describe the subsequent OAC switch, this could be added in an additional ‘Comments’ column.
- Given how descriptive Table 2 is, lines 194-234 are overly detailed, repeating the information in the table. A brief summary of any obvious trends (or absence of trends) could be provided.
- Lines 247-252: Consider rewording to “There was significant variability in dose appropriateness when prescribing DOACs but no clear association between dose appropriateness and bleeding. Patients who experienced a bleed also had a number of other factors predisposing to bleeding, including a high proportion of supratherapeutic INRs or concomitant antiplatelet therapy.”
- Line 261: Consider rewording “it is not surprising that of the DOAC options, it’s (sic) use is high” to more academic language.
- Line 264: Consider “… in those”
- Discussion: As above, this is overly long and repetitive, and needs to be tightened up significantly. Reference to other papers should be focussed around the desired message, without repeating too much of the characteristics of the study. While it is interesting to read how these data have been utilised in the authors’ practice, this could be summarised in one or two sentences.
Author Response
Reviewer #1 Report Round 2, Comments/Author Response:
- Line 65: Please add ‘in’ between ‘use’ and ‘AF’ – i.e. “Despite the lack of guidance from US guidelines for DOAC use in AF patients receiving HD…"
Author Response:
Done. Inserted ‘in’ between ‘use’ and ‘AF’
- Lines 69-79 remain overly detailed and repetitive. Time period, methods, results, etc. of other studies should only be described in general terms, unless the details are required for comparative purposes. This also applies to the Discussion.
Author Response:
I deleted Line 69-72. Line 72-75 starting with “The study found an increase in prescribing…” as I believe that is germaine to the awareness of utilizing these agents in the dialysis population right after their FDA approval, even though this population was excluded from the landmark AF trials and outcome information on their safety/efficacy was unknown. Therefore, I also currently kept Line 75-77 starting with “Unfortunately, this study found an increase in the risk of hospitalization or death from bleeding…” and changed it to “This study found an increase in the risk of….” as I believe this is an important finding from their study illustrating the high risk of using these particular DOACs in the HD population. Also, for Line 78-79, I changed “Therefore, at the time of this study, no data was available…” to “Therefore, at the time of our study, no data was available evaluating the safety or outcomes using apixaban or edoxaban in the dialysis population.”
- Lines 110-112: Thank you for including relevant definitions in your Methods. Please note that, unless the patients’ INR results were all spaced evenly apart in terms of timing, what is reported is not TIME in therapeutic range (which is typically calculated using Rosendaal’s interpolation method), but the percentage of therapeutic/subtherapeutic/supratherapeutic INRs.
Author Response:
I will make this change to reflect percentage of values reported that were therapeutic/subtherapeutic/supratherapeutic to get the TR%, SubR%, SupraR%, respectively. Thus, in the Methods section and Table 2 column corresponding to these percentages, this will be clarified. The Methods section, Line 110-112 were revised to read, “For each patient the percent of values in subtherapeutic range (SubR%), therapeutic range (TR%), and supratherapeutic range (SupraR%) were calculated based on the number of INR values that were either subtherapeutic, therapeutic, or supratherapeutic divided by the total number of INRs documented for that patient, multiplied by 100 to get the SubR%, TR%, and SupraR%, respectively.” If use of TTR%, TSubR%, or TSupraR% were used in the Results or Discussion section, these were revised to TR%, SubR%, or SupraR% throughout.
- Lines 120-126 are not required, having been defined in Appendix A.
Author Response:
The Lines 120-124 were deleted, however I left Lines 124-126 which define “NDA, dose high” and “NDA, dose reduced” as those terms are used in Figure 1, Table 2, and Results section. If the reviewers/editors feel that this is clear and implied with “NDA, dose reduced” or “NDA, dose high” and delete the further definitions of NDA, then that is allowable. I just left those lines in for now.
- Lines 130-133: The definitions of major bleeding provided are somewhat contradictory. Was the threshold ‘bleeding causing a reduction in the hemoglobin level of at least 2 g/dL’ or ‘5g/dL’? And was it bleeding ‘leading to a transfusion of at least 2 units of blood’ or ‘at least 4 units of blood’? Please clarify.
Author Response:
The clinical cutoffs for determining major bleeding were conservative, thus identifying it when bleeding caused a reduction in the Hgb level of at least 2 g/dL and bleeding that led to a transfusion of at least 2 units of blood. Will revise Lines 130-133 to clarify this.
- Lines 159-171: Figure 1 is highly descriptive; the results in the text are largely repeated from the figure and could be further summarised or deleted.
Author Response:
Lines 161-171 were revised to summarize the information. Information that was deleted from this paragraph included highlighting prescribing with “dose appropriateness”, since this is discussed in its own section Line 194-217.
- Figure 1: How is ‘Med stopped’ DA? The Figure 1 title block (lines 178-9) is repetitive.
Under the apixaban 2.5 initial DOAC, then “Med stopped, DA” is Row #26 patient where the correct DOAC dose of apixaban 2.5 mg BID and patient experienced a CRNMB/minor bleed that was a GI bleed and the med was stopped. Patient #26 was referred to in the Results section Line 203-204, however it did not explicitly state for Patient #26 that the med was stopped. This will be clarified in the text and I will take out the box currently showing “Med Stopped, DA” under the apixaban 2.5mg arm in Figure 1. Under the warfarin- then apixaban 2.5 arm, then “Med stopped, NDA” is Row #23 patient where low dose apixaban 2.5 mg BID was prescribed when the patient did not meet dose reduction criteria and patient experienced a major GI bleed and the med was stopped. This is described in the Results section Line 217, this I will take out the box currently showing “Med Stopped, NDA” from Figure 1. The revised Figure 1 will be sent to you as an attachment with the title of the document “Figure1_InitialOACPrescribedDA_FinalV2”.
Figure 1 title block Line 178-180 was deleted to decrease repetitiveness. Line 180 information defining DA and NDA was retained.
- Table 2 is well constructed and highly descriptive. Given that column 2 is ‘OAC at time of bleed’, why do some lines (e.g. Patients 21, 22, etc.) have rows where there was no bleed? If the aim is to describe the subsequent OAC switch, this could be added in an additional ‘Comments’ column.
Author Response:
Patient #19 through #24 were patients who were on different OACs throughout the study observation period. For example, patient #21 was initially on warfarin for 1 month and experienced a major GI bleed, no minor bleed and INRs documented during that time are reported as TR%, SupraTR%; then switched to apixaban 2.5mg BID (that was not dose appropriately as lower dose was used when the patient did not meet dose reduction criteria) for 2 months and did not experience any bleed. The aim of the table is not just to describe OAC switch, it is to illustrate duration of OAC therapy before switch, concomitant antiplatelet therapy during a bleed (or no bleed), appropriateness of OAC dose, and the type of bleed occurring while on which OAC therapy. I would prefer to leave Table 2 as it stands, and I believe it illustrates the complexity of prescribing of these agents; when large claims data cohort studies would not capture this information as they likely stay with the first OAC prescribed or perhaps an OAC prescribed the majority of the time for a patient. Figure 1 illustrates that complexity of switching in prescribing therapy, and Table 2 illustrates potential factors beyond OAC switch or dose change that could have possibly impacted development of a bleed.
- Given how descriptive Table 2 is, lines 194-234 are overly detailed, repeating the information in the table. A brief summary of any obvious trends (or absence of trends) could be provided.
Author Response:
The information in Lines 194-234 was revised to summarize a trend (or lack thereof) for appropriateness of DOAC dose or patient characteristics associated with developing a bleed.
- Lines 247-252: Consider rewording to “There was significant variability in dose appropriateness when prescribing DOACs but no clear association between dose appropriateness and bleeding. Patients who experienced a bleed also had a number of other factors predisposing to bleeding, including a high proportion of supratherapeutic INRs or concomitant antiplatelet therapy.”
Author Response:
This was revised as recommended.
- Line 261: Consider rewording “it is not surprising that of the DOAC options, it’s (sic) use is high” to more academic language.
Author Response:
This sentence was revised.
- Line 264: Consider “… in those”
Author Response:
This sentence was revised.
- Discussion: As above, this is overly long and repetitive, and needs to be tightened up significantly. Reference to other papers should be focussed around the desired message, without repeating too much of the characteristics of the study. While it is interesting to read how these data have been utilised in the authors’ practice, this could be summarised in one or two sentences.
Author Response:
The discussion section was revised to reduce repetition of specifics from the study and minimized information from related studies to focus on the desired message. Information regarding impact of study on the practice site was summarized.
Author Attention to Editors re: Reference numbering for Editors to be aware of:
It was noted during review of Track Changes with “No Markup” that there were some shifts in autonumbering of References. Therefore, the version I received had a “blank/empty” citation for Reference #19 and Xarelto package insert had moved to Reference #20. Similarly, the version I received had a “blank/empty” citation for References #29-31 and Yao et al had moved to Reference #32. Therefore, I moved the citations to the appropriate autonumbering starting with #19 should be Xarelto PI ending with #27 as Wang et al. Then #28 is now Reed et al (as I moved where the section where this was referenced to in the Discussion section up. Then #29 is now Yao et al. #30 is Pokorney et al. #30 Zyl et al was deleted. Reinecke et al. was #32 and is now #31. ClinicalTrials.gov NCT03563937 was #33 and it was deleted. Coleman et al was deleted.